# Landscape conservation and orchard management influence carob tree yield through changes in pollinator communities

**Carmelo Gómez-Martínez**[1], **Miguel A. González-Estévez**[1],
**Indradatta deCastro-Arrazola**[1], **Peter Unglaub**[1], **Amparo Lázaro**[1]*

**1** Global Change Research Group, Mediterranean Institute for Advanced Studies, Esporles, Balearic Islands, Spain

* amparo.lazaro@imedea.uib-csic.es

## Abstract

Worldwide pollinator declines are a major problem for agricultural production. However, understanding how landscape characteristics and local management influence crop production through its pollinators is still a challenge. The carob tree (*Ceratonia siliqua*) is a pollinator-dependent Mediterranean crop of high economic importance in food and pharmaceutical industries. To understand how crop production can be enhanced in a sustainable manner, we evaluated the effects of landscape (habitat loss) and orchard local management (farming system: conventional vs. ecological; male-to-female ratio) on pollinator communities and crop production using data on 20 carob tree orchards across Mallorca Island (Spain). We found that orchards surrounded by a greater proportion of natural landcover received more visits by wild bees and butterflies and fewer by honeybees. Overall pollinator abundance was slightly higher in ecological than conventional orchards, but the difference was not significant. High male-to-female ratio enhanced overall pollinator abundance and shaped pollinator composition, by increasing hoverfly abundance and decreasing wasp and fly abundance. Male-to-female ratio showed hump-shaped relationships with fruit and seed production per female tree (peak at 0.7 males/female), although this quadratic relationship was lost when the most male-biased orchards were removed from the analyses. Total orchard production maximized with 25-30% of males. Seed weight (farmer's highest economic value) increased in conserved landscapes where wild pollinators prevailed, and with overall pollinator abundance; however, it decreased with male-to-female ratio, likely due to seed number-size trade-offs. Management strategies to enhance carob production may optimize sex ratios and favor wild pollinators by preserving natural landscapes.

## Introduction

The ongoing decline of wild insects [1–3] poses a threat to the essential ecosystem service of pollination [4]. Land-use changes are considered one of the main factors driving such pollinator declines, as they lead to the homogenization of landscapes [5] and communities [6], affecting habitat and resource availability for pollinators [7], and disrupting plant-pollinator

**Data availability statement:** Data are available at Zenodo. DOI: 10.5281/zenodo.13939479

**Funding:** This study was supported by the project CGL2017-89254-R, financed by the Spanish Ministry of Science and Innovation, FEDER Funds and the Spanish State Research Agency, and by the project PRPPID2020-117863RB-I00, financed by the Spanish Ministry of Science and Innovation and the Spanish State Research Agency (MCIN/AEI/10.13039/501100011033). CGM was supported by a FPI predoctoral contract financed by the Spanish Ministry of Economy and Competitiveness, the Spanish Research Agency, and European Social Funds (FPI PRE2018-083185, Call 2018). The funders had no role in study design, data collection and analysis, decision to publish, or preparation of the manuscript.

**Competing interests:** The authors have declared that no competing interests exist.

interactions [8]. This pollinator loss is especially worrisome for the maintenance of agricultural production, because 75% of cultivated plants and 35% of global production depend on insect pollination [4]. Although managed honeybees are commonly used to pollinate crops, wild insects are often more effective pollinators [9–12], and pollination by wild insects is known to increase crop production quantity [4,10,13] and quality [14] and, therefore, farmers' profits [15].

Several studies have shown that habitat loss has a negative effect on crop production due to decreased pollinator visits [13,16–19]. A larger amount of natural habitat in the landscape allows a richer insect interchange between natural habitats and crops [20], while landscapes dominated by crops may have abundant but less diverse pollinator communities [21,22]. The local management of orchards may also have additional effects on crop pollinators and production. A higher pollinator diversity and pollination success has been found in orchards with a living ground cover [16,23], as well as in ecological crops (also called organic crops) [24–27], which do not use synthetic agrochemicals, antibiotics or genetically modified organisms, and maintain natural habitats surrounding the fields. Besides, the negative effects of non-ecological management on pollinator abundance and richness may be stronger in simplified than in complex landscapes, because complex landscapes already have high diversity of pollinators [28–30].

The effects of pollinator loss on crops might also depend on the crop breeding system and its dependence on pollinators [31–33]. Dioecious crops, which rely on pollinator visitation to both sexes for sexual reproduction, might be particularly vulnerable to overall pollinator decreases [34], as well as to the loss of the most efficient species [35]. For this type of crop, optimizing the sex ratio during orchard design could be critical [36]. An optimized sex ratio might be important to maximize crop production because more females imply more individuals producing seeds while enough males are needed to provide suitable levels of pollen to fertilize female flowers. In addition, an optimized sex ratio might also help control for density-dependent processes of facilitation-competition between sexes [37], as males could help attract pollinators to females when they are at low-to-medium abundances, but compete with females for pollinator visits when they are at high abundances. Indeed, it has been reported that a too high proportion of males might be counterproductive for pollinator visits to females in cases where males are more attractive to pollinators [38–40].

The carob tree (*Ceratonia siliqua* L.) is a crop belonging to the family Fabaceae, widely cultivated in countries of the Mediterranean Basin, and other areas with Mediterranean climate [41,42]. The carob pods, seeds, and gum have many nutritional, biochemical, and clinical applications, with the seeds being the most profitable part of carob fruits [43]. The applications include livestock feeding, food, and diverse pharmaceutical uses [44–47]. Spain is the largest producer worldwide (60-80 thousand tons produced per year) [48], and the Balearic Islands contribute around a third of the Spanish production. Carob tree is a polygamo-trioecious species (with female, male, and hermaphroditic plants, and plants having bisexual and male flowers and others having bisexual and female flowers) [41,49], but it mostly behaves as dioecious in nature [50–52]. The flowers are inconspicuous, with no developed corolla, and grow in racemes in old branches (S1 Fig in Supporting information). The fruits are dark-brown pods with hard seeds and a sweet pulp and take almost a year to mature [53]. Carob trees are highly dependent on animal pollination for reproduction, with insects increasing pod production from up to 16 times compared to wind-pollination [54]. Although female trees produce nectar, male trees produce both nectar and pollen, and are more fragrant and tend to have a larger flower display, which could serve as attractant for pollinators [41,55]. However, as male trees do not produce fruits, farmers increase female abundance by grafting female branches in male rootstocks and, as a consequence, male tree abundance is usually much

lower than the abundance of female trees, or even null, which might lead to spatial isolation of female trees, hindering pollination [56]. To our knowledge, there are no studies to date that have evaluated how the landscape context and local management of orchards influence carob tree yields through effects on their pollinators. Understanding this is important to alleviate the potential negative effects of landscape disturbances on wild pollinator communities, and to enhance crop production in a sustainable manner. With this aim, we collected data on pollinator visitation to carob tree flowers and fruit and seed production in 20 carob tree orchards along a gradient of percentage of natural area in the landscape on Mallorca Island (Spain). Particularly, we analyzed how the percentage of surrounding natural habitat (Fig 1A), the farming system (conventional vs. ecological; Fig 1B) and the ratio of male-to-female trees in the orchards (Fig 1C) influenced: 1) the composition and abundance of pollinators visiting carob orchards; 2) the production of carob fruits and seeds, and 3) the quality of carob seeds, estimated as seed weight. We expected that orchards within conserved landscapes and subjected to ecological farming would hold a higher pollinator abundance and would have diverse communities composed of wild pollinators that enhance carob production; and that the negative effect of habitat loss on crop production would be less pronounced in ecological farms. Lastly, we expected an increase in pollinator visitation rates with the increase in male trees in the orchards, whereas a hump-shaped relationship between the proportion of male trees and crop production, related to density-dependent competitive processes between sexes.

## Materials and methods

### Study orchards

We selected 20 carob tree orchards (study orchards, hereafter) across Mallorca Island, in the Balearic Archipelago, Spain (S2 Fig in Supporting information), after obtaining permission

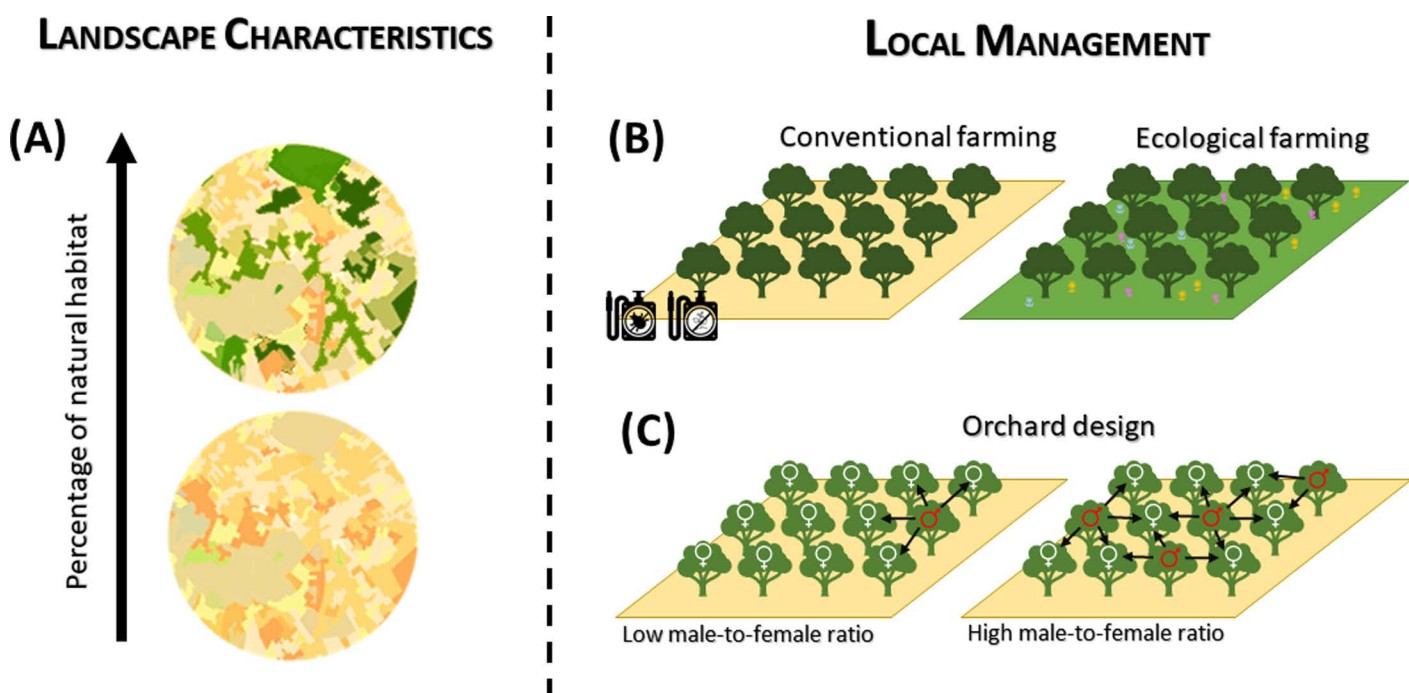

**Fig 1. Conceptual diagram with the variables potentially influencing carob tree production.** (A) The percentage of natural habitats in the landscape surrounding the orchards; (B) The farming system (conventional vs ecological Farming); (C) The proportion of carob male trees per female tree.

from the owners to work in their properties. The study orchards were selected along a gradient of natural habitat loss (see "Landscape characterization" section), and separated from each other by a minimum distance of 1.50 km from the center of the orchards (mean ± SE distance among study orchards: 32.07 ± 1.42 km). The study orchards were located between 43 to 180 m.a.s.l. and had a mean size of 4.41 ± 1.09 ha, and a mean tree density of 66 ± 8 trees/ha (S1 Table in Supporting information). All the study orchards, and both female and male flowers, flowered during the same period (from beginning of October to mid-end November); however, it cannot be totally discarded that study sites include different cultivars, as this information was unknown by the farmers. Mean temperatures for October and November in Mallorca Island were 19.7ºC and 13.5ºC respectively in 2019, and 17.2ºC and 15.6ºC respectively in 2020. The farmers of 12 of the study orchards reported the presence of honeybee hives on their own property or in adjoining parcels (S1 Table ). During the flowering season of carob trees in Mallorca, hives hold between 20-40 thousand individuals (personal communication from local beekeepers). Honeybees are normally the most common pollinator in extensive crops in Mallorca that flower in winter and autumn, as almond and carob trees, independently of whether the owners have honeybee hives in their properties or not [16]. This is because these traditional orchards are usually small and embedded within a heterogeneous landscape that includes crops and natural areas, and because wild pollinators are still scarce during these periods of the year, and honeybees are abundant and able to travel several kilometers for foraging [57,58].

## Landscape characterization

We used the last update of the SIOSE AR database (Spanish acronym for Soil Occupation Information System of Spain - High Resolution) [59] to describe the characteristics of the landscape surrounding the study orchards within a buffer zone of 1-km radius from their center. We used this radius because most pollinators have foraging ranges not larger than 1-km radius e.g., [60,61]. We had a total of 43 land-cover classes in the buffer zones, 7 of which corresponded to natural (mix of shrub and pasture, mix of wood and pasture, woodland, perennial and coniferous forest, and shrubland) or semi-natural habitats (pastures, 0 to 0.34% of the buffer area in each site). The remaining 36 land-cover classes corresponded to crops (mainly non-citric fruit trees and herbs; 13 classes), artificial (17 classes) and water bodies (6 classes). With these data we estimated the percentage of natural and semi-natural areas from the total area within the 1-km buffers (% natural habitat, hereafter; S1 Table ). The percentage of natural areas was negatively correlated to the percentage of crops in the landscape (r = -0.85, t = -6.84, n = 20, p < 0.0001).

## Local management of orchards

To study potential effects of the local management on the pollinator community and production of carob tree orchards, we considered the farming system (conventional vs. ecological), and the male-to-female ratio (number of male trees per female tree). We classified the study orchards as conventional or ecological according to the European regulation Nº 2018/848, which in the Balearic Islands is managed and applied by the CBPAE (acronym in Catalan for the Council for Ecological Agricultural Production of Balearic Islands). Seven of the study orchards were certified as ecological farms, while the remaining 13 were under conventional management (S1 Table ). According to the CBPAE, ecological farming does not make use of synthetic agrochemicals, maintains non-cultivated areas and natural habitats to promote biodiversity, must be sustainable, and does not use genetically modified organisms (GMO). Regarding the male-to-female ratio, we registered the sex of every carob tree in each study

orchard and divided the number of males by the number of females to get a ratio of males to female (male-to-female ratio, hereafter; S1 Table ). The male-to-female ratio in carob orchards of Mallorca is generally low, as farmers obtain female carob trees by grafting female branches in male rootstocks to increase the number of female trees that produce fruits. Hermaphrodite trees were absent in 11 of the 20 study orchards and scarce in the rest (from 1 to 8%), except for one study orchard where most of the individuals (95.7%) were hermaphrodite (Son Cotoner; S2 Fig ). For this reason, and to simplify the male-to-female ratio estimate in the study orchards, we counted the hermaphrodite trees both as males and females.

## Pollinator visitation

We recorded visits of pollinators to carob tree flowers during two flowering seasons (2019 and 2020), from the beginning of October to the beginning of November, covering the flowering period of this crop in Mallorca. Each study orchard was sampled five days during the flowering period (approximately once per week). Three observers recorded pollinator visits to carob flowers while walking slowly along tree lines for one hour each sampling day at each study orchard. Pollinator censuses were carried out between 10.00h and 16.00h, in sunny days without wind, to ensure pollinator activity. We considered a visit of a pollinator to a flower when a pollinator was observed contacting the reproductive parts of the flower. We noted if the pollinator contacted a male, female, or hermaphroditic flower. As flowers are very close to each other in a raceme, we counted as one interaction a visit of a pollinator to a raceme, although they might contact several flowers. As flower racemes of carob trees are caulogenous (emerging only from the old branches) [62], collecting visiting insects between branches and leaves with a hand-net becomes difficult. Therefore, during censuses, we categorized each visitor into eight functional groups [63]: butterflies, hoverflies, bee flies, other flies (mainly muscoids and small acalyptrate flies), ants, wasps, wild bees, and honeybees (*Apis mellifera*). Nevertheless, we captured the insects when possible for their identification in the lab if their identification to the species level in the field was unfeasible (pausing the watch during manipulation to standardize the time spent per study orchard). With the visitation data, we calculated (1) the pollinator abundance by functional group, as the total number of individuals of each functional group recorded in each study orchard each year, and also (2) the overall pollinator abundance, as the total number of individuals recorded in each study orchard each year.

## Carob tree production

In both study years, at the beginning of the flowering period (beginning of October in 2019 and 2020), we randomly selected 20 female trees in each study orchard and randomly marked one of its branches to count the number of inflorescences and inflorescence buds (37.85 ± 1.44 mean (SE) inflorescences per marked female tree). We then counted the number of flowers and flower buds in five random inflorescences of each marked branch and estimated the total number of flowers in the marked branch as the average number of flowers per inflorescence multiplied by the number of inflorescences in the branch (969.55 ± 33.91 flowers per marked female tree). After fructification (August in 2020 and 2021) and before harvest, we counted the number of aborted and developed fruits in each marked branch, and estimated fruit set per branch as the number of developed fruits divided by the total number of flowers. We also collected five fruits per marked branch to study seed production and weight in the lab, by counting the number of developed and aborted seeds separately for each of the five fruits per tree, and weighing all the developed seeds per fruit together.

With all these data, we then calculated (1) *Fruit production per female tree*, as the fruits produced by 1000 flowers in a tree (fruit set x 1000); (2) *Seed production per female tree*, as the seeds

produced by 1000 flowers in a tree (seeds/fruit x fruit production per tree); and (3) *Seed weight*, as the weight (in grams) of one seed, calculated for each fruit as the total mass of developed seeds divided by the number of developed seeds in that fruit. Fruit and seed production per tree were estimated per 1000 flowers because fruit set is very low in carob trees, but the number of fruits per tree may be relatively high due to the large number of flowers produced [53].

To shed light on the optimum male-to-female ratio in carob tree orchards, we additionally estimated the percentage of males in the orchards (males/100 trees) that maximized total fruit and seed production, and the total weight of seed production at the orchard level (hereafter orchard-level fruit production, orchard-level seed production, and orchard-level total seed mass, respectively). We estimated orchard-level fruit and seed production by multiplying the production of a female tree by the percentage of females in the orchard. Orchard-level total seed mass was estimated as the orchard-level seed production multiplied by the estimated seed weight at different male-to-female ratios (see below for results of GLMMs). Note that fruit and seed production per tree was estimated per 1000 flowers.

## Statistical analysis

We performed all the statistical analyses in R v.4.2.2 [64]. To study how pollinator community composition changed with the percentage of natural habitat surrounding the study orchards, the male-to-female ratio and the farming system (conventional vs ecological), we ran a Canonical Correspondence Analysis (CCA; R-package vegan v.2.6.4) [65]. We also included the year as predictive variable, as pollinator communities might change between study years, and tree density (number of trees per hectare; S1 Table ), as this variable might also influence the results. The response variables were the abundance (number of visits) per study orchard and year of each pollinator group: honeybees, wild bees, wasps, ants, hoverflies, other flies, and butterflies. As the number of beetles and bee flies was extremely low (see Results section), we removed these two pollinator groups from the analyses. To assess the relationship between the response variables and the predictors, we first evaluated the significance of the whole ordination by performing Monte Carlo permutations [65,66]. The significance of the model was determined by comparing the observed statistics against a null distribution generated from 9999 random permutations of the community data, while keeping the predictors fixed. To determine the significance of each predictive variable, we performed conditional permutation tests. In these tests, the significance of each variable was evaluated by randomizing the variable to test while keeping the rest of predictors fixed. The observed statistics for each variable were compared against a null distribution generated from 9999 random permutations [65,66].

To evaluate how overall pollinator abundance was related to the surrounding landscape and the local management of the study orchards (male-to-female ratio and farming system), we fitted a generalized linear mixed model (GLMM; R-package lme4 v.1.1.32) [67], with the study orchard as a random factor to control for pseudoreplication. Thus, after checking for the absence of collinearity (VIF values < 3) [68], we included the percentage of natural habitat, the male-to-female ratio, and the farming system (conventional vs. ecological) as predictor variables in the full model; in addition, we also included tree density and the sampling year to control for any potential effect that these variables could have on the results. As previous studies showed a potential synergic effect between landscape disturbance and local management [28,29], we also included the interaction between the percentage of natural habitat and the farming system in the full model. To evaluate how carob trees' production was related to the landscape surrounding the study orchards and the local management, we fitted separated GLMMs with fruit production per female tree, seed production per female tree, and seed weight as response variables. The predictor variables were the same as for the GLMM models of pollinator visits, but we also added total pollinator abundance and the first axis of the

CCA as predictor variables to assess whether the abundance and composition of pollinators in the study orchards directly influenced carob production. We did not include the second axis of the CCA as it was not significant (see Results section). As we expected a hump-shaped relationship between carob production and the male-to-female ratio due to density-dependent competence/facilitation processes [37,69], we included the male-to-female ratio both as linear and quadratic terms, first by standardizing the linear term to $\bar{x} = 0$ and $\sigma = 1$, and then calculating its quadratic form [70] to avoid collinearity [68]. Both study orchard and tree nested within study orchard were included as random factors in these models. Due to overdispersion [68], we used a negative binomial distribution (link log) for the models of pollinator abundance, and fruit and seed production per female tree. A gamma distribution (link log) was used for the model of seed weight as it did not fulfill the assumptions of normality even after log-transformation (R-package nortest v.1.0.4) [71]. We conducted multi-model inference based on AICc (R-package MuMIn v.1.47.5) [72]. For each full model, we constructed sub-models containing different combinations of predictor variables, limiting the maximum number of predictor variables to four in each model, to avoid over-parametrization due to sample size. We then calculated unweighted averages using the sub-models with $\Delta$AICc $\leq 2$ (function model.avg in R-package MuMIn v.1.47.5) [72], and show the results of full model averaging in the text. Residual inspection indicated adequate model fits. Plots were drawn using the R-package ggplot2 [73], except the CCA that was drawn using the function plot.cca in R-package vegan v.2.6.4. [65].

To ensure that our statistical analyses were adequate, we run Moran's I tests using the function moran.mc in R-package spdep v.1.3.6 [74], to assess whether there was spatial autocorrelation in our data (pollinator abundance, fruit and seed production and seed weight). The results of these tests indicated that there was no spatial autocorrelation in the data (S2 Table in Supplementary Material) and, therefore, our procedure was appropriate. In addition, to evaluate the extent to which our results could be influenced by orchards with sex-ratios particularly different, we repeated all the models excluding the study site with the highest number of hermaphrodite plants ("Son Cotoner") and the study site with the highest percentage of males ("Ses Figueres").

## Results

We registered a total of 8789 pollinator visits to carob trees (4014 in 2019 and 4775 in 2020). More than half of the visits (ca. 57%; 5017 visits) were conducted by honeybees (1857 visits in 2019 and 3160 in 2020), followed by wasps (1527 visits; 763 in 2019 and 764 in 2020), muscoids and small acalyptrate flies (other flies, 917; 470 in 2019 and 447 in 2020), hoverflies (561; 322 in 2019 and 239 in 2020), wild bees (382; 326 in 2019 and 56 in 2020), ants (241; 156 in 2019 and 85 in 2020), butterflies (119; 112 in 2019 and 7 in 2020), beetles (14; 11 in 2019 and 3 in 2020) and bee flies (11; 5 in 2019 and 6 in 2020). S3 Table in Supporting information shows detailed information on the number of visits per pollinator group, study orchard and year. Although all the pollinator groups visited both sexes, there were significant differences in pollinator composition between them (PERMANOVA: $R^2 = 0.127$; $F = 5.392$; $p = 0.006$), as male flowers were mainly visited by honeybees (ca. 72% of visits), while visits to female flowers were more diverse (honeybees: 48%; wasps: 22%, other flies: 15%; S4 Table in Supporting information). We identified 45 species of wild insects visiting carob tree flowers (S5 Table in Supporting information), belonging to ants (6), wild bees (7), wasps (3), other flies (muscoids and small acalyptrate flies) (15), hoverflies (8), bee flies (1), butterflies (4) and beetles (1). The number of fruits per female tree varied between 8.8 and 45.2 fruits in one thousand flowers, depending on the study orchard (mean ± SE: 26.7 ± 2.5 fruits in one thousand flowers). The number of seeds per female tree varied between 88.6 and 558.3 seeds in one thousand flowers,

depending on the study orchard (286.6 ± 30.35 seeds in one thousand flowers). Seed weight varied between 0.09 and 0.33 grams depending on the study orchard (0.19 ± 0.001 grams per seed).

## Pollinator community composition

The CCA indicated significant relationships between the abundance of the pollinator groups, the landscape characteristics and local management of the orchards ($F_{4,35}$ = 3.49, p = 0.0004; Fig 2A). The cumulative percentage of variance explained by the first axis was 70.8%, while the two first axes explained 90.3% of the total variance. The percentage of natural habitat, the farming system and the year varied along the first ordination axis, which was significant (CCA1, $F_{1,35}$ = 9.90, p = 0.0003). The male-to-female ratio varied along the second axis; however, this second axis was not significant (CCA2, $F_{1,35}$ = 2.71, p = 0.462). The predictor variables that significantly influenced the pollinator community composition were the percentage of natural habitat ($F_{1,35}$ = 2.99, p = 0.047), the male-to-female ratio ($F_{1,35}$ = 3.10, p = 0.042) and the year ($F$ = 6.47, p = 0.0008). Higher percentage of natural habitat was mainly related to an increase in the abundance of wild bees, but also of ants and butterflies, whereas more disturbed landscapes (lower percentage of natural habitats) were related to an increase in the abundance of managed honeybees. Higher male-to-female ratio was positively related to increased hoverfly abundance, while other groups as ants, wasps and other flies were related to low male-to-female ratios. Lastly, the abundance of wild bees and butterflies was higher in 2019 than in 2020, while the other pollinator groups did not significantly change their abundances between years. The results did not change when we excluded from the analysis the study site with the highest number of hermaphrodite plants and the study site with the highest percentage of males (S6 Table in Supporting information).

## Overall pollinator abundance

The results of model averaging indicated that higher male-to-female ratios were positively related to the overall abundance of pollinators in the orchards (Table 1A, Fig 2B). Although fields subjected to ecological farming showed slightly higher values of overall pollinator abundance per study site and year than those under conventional farming (mean ± SE: 246.86 ± 29.80 and 205.11 ± 23.97, for ecological and conventional farming, respectively), the difference was not significant (Table 1A). The results of this analysis did not change when we excluded the study site with the highest number of hermaphrodite plants and the study site with the highest proportion of male trees (S7A Table in Supporting information).

## Carob tree production

The results of model averaging for fruit and seed production per female tree showed similar results. Both models indicated that fruit and seed production per female tree increased with male-to-female ratio until a maximum at values of 0.6-0.7 males/female, after which higher ratio values decreased the production per female tree (Table 1B and C, Fig 3A and B). With these data, we estimated that orchard-level fruit and seed production was maximized with 25-30% of male trees in the orchards (Fig 3C and D). However, the quadratic effect was lost when we excluded from the analyses the study site with the highest number of hermaphrodite plants and the most male-biased study site (S7B Table in Supporting information), which were the two localities with highest sex ratios.

Regarding seed weight, we found a negative significant relationship between seed weight and the first axis of the CCA (CCA1, related to the percentage of natural habitat in the landscape and the abundance of wild bees vs. honeybees, Table 1D, Fig 4A). Thus, an increase

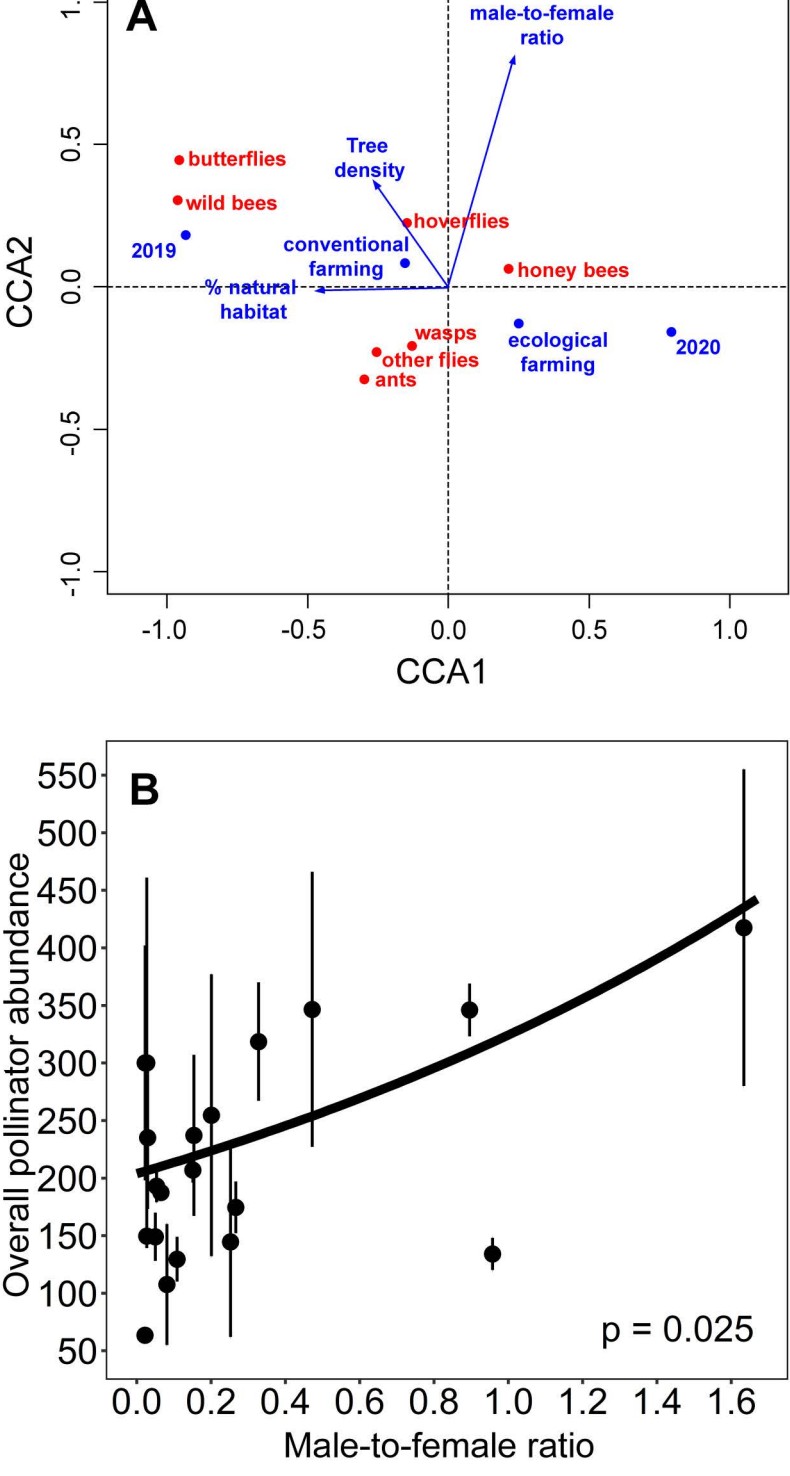

**Fig 2. Effects of landscape and local management on pollinator community composition and overall pollinator abundance in carob orchards.** (A) Canonical Correspondence Analysis (CCA) showing the relationships between the percentage of natural habitat, male-to-female ratio, farming system and year (blue arrows and dots) and the pollinator groups (red dots). Short distances between the pollinator groups and the predictor variables in the ordination indicate high association between them. Percentage of natural habitat ($F$ = 2.99, **p** = 0.045); male-to-female ratio ($F$ = 3.10, **p** = 0.042); year ($F$ = 6.47, **p** = 0.0008). (B) Relationship between the male-to-female ratio and overall pollinator abundance. Lines represent the estimates of the best model, the dots represent the average of total visits per year and study orchard, and the vertical lines the standard errors.

**Table 1. Model-averaged coefficients, significance, and relative importance (RI) of the predictor variables for the GLMMs to evaluate effects on: (A) overall pollinator abundance; (B) fruit production per female tree, (C) seed production per female tree; and (D) seed weight. Farming system refers to ecological vs. conventional practices; CCA1 refers to the first axis of the CCA and depicts the relationship between the % of natural habitats in the landscape and the composition of the pollinator community (increase abundance of wild bees with the increase in natural habitats); Male-to-female ratio and Male-to-female ratio$^2$ indicate linear and quadratic terms, respectively. Significant *p*-values are marked in bold. RI was estimated by summing the Akaike weights over all the models where each explanatory variable appeared, and highlights the importance of these variables in the models [75].**

| Model | Predictor | Estimate | Std. error | z value | p | RI |
|---|---|---|---|---|---|---|
| (A) Overall pollinator abundance | (intercept) | 4.922 | 0.258 | 18.710 | **< 0.0001** | |
| | Male-to-female ratio | 0.464 | 0.200 | 2.237 | **0.025** | **1.00** |
| | Tree density | 0.003 | 0.003 | 1.073 | 0.283 | 0.69 |
| | Year | 0.063 | 0.118 | 0.524 | 0.600 | 0.32 |
| | Farming system | 0.154 | 0.196 | 0.776 | 0.438 | 0.50 |
| | % Natural habitat | −0.001 | 0.002 | 0.252 | 0.801 | 0.16 |
| (B) Fruit production per female tree | (Intercept) | 3.555 | 0.213 | 16.653 | **< 0.0001** | |
| | Male-to-female ratio$^2$ | −1.371 | 0.461 | 2.967 | **0.003** | **1.00** |
| | Male-to-female ratio | 1.018 | 0.430 | 2.362 | **0.018** | **1.00** |
| | Year | −0.084 | 0.095 | 0.852 | 0.378 | 0.64 |
| | Overall pollinator abundance | −0.0004 | 0.001 | 0.742 | 0.458 | 0.51 |
| | Farming system | 0.027 | 0.099 | 0.294 | 0.776 | 0.19 |
| | CCA1 | −0.003 | 0.031 | 0.102 | 0.919 | 0.15 |
| | Tree density | −0.001 | 0.0007 | 0.093 | 0.926 | 0.06 |
| | % Natural habitat | −0.002 | 0.002 | 0.117 | 0.907 | 0.10 |
| (C) Seed production per female tree | (Intercept) | 5.947 | 0.231 | 25.685 | **< 0.0001** | |
| | Male-to-female ratio$^2$ | −1.320 | 0.489 | 2.698 | **0.007** | **1.00** |
| | Male-to-female ratio | 0.919 | 0.455 | 2.176 | **0.029** | **1.00** |
| | Year | −0.178 | 0.121 | 1.471 | 0.141 | 0.91 |
| | Overall pollinator abundance | −0.0003 | 0.001 | 0.546 | 0.585 | 0.36 |
| | CCA1 | 0.015 | 0.046 | 0.316 | 0.752 | 0.20 |
| | Farming system | 0.036 | 0.121 | 0.322 | 0.747 | 0.20 |
| | % Natural habitat | −0.0001 | 0.002 | 0.103 | 0.918 | 0.08 |
| | Tree density | −0.0001 | 0.0008 | 0.067 | 0.947 | 0.08 |
| (D) Seed weight | (Intercept) | −1.673 | 0.033 | 50.965 | **< 0.0001** | |
| | CCA1 | −0.020 | 0.004 | 4.568 | **< 0.0001** | **1.00** |
| | Overall pollinator abundance | 0.0002 | 0.000 | 3.545 | **0.0004** | **1.00** |
| | Male-to-female ratio | −0.114 | 0.045 | 2.504 | **0.012** | **1.00** |
| | Tree density | −0.0003 | 0.0001 | 0.711 | 0.477 | 0.51 |
| | Year | −0.009 | 0.009 | 1.079 | 0.281 | 0.23 |
| | % Natural habitat | −0.0002 | 0.001 | 0.256 | 0.798 | 0.17 |
| | Male-to-female ratio$^2$ | 0.006 | 0.034 | 0.188 | 0.851 | 0.15 |
| | Farming system | −0.001 | 0.009 | 0.120 | 0.904 | 0.07 |

in the abundance of wild bees in more conserved landscapes was positively related to seed weight (Table 1D, Fig 4A). The model also showed a positive significant relationship between seed weight and overall pollinator abundance (Table 1D, Fig 4B), and a negative relationship between seed weight and the male-to-female ratio (Table 1D, Fig 4C). Same as for fruit and seed production, we estimated that orchard-level total seed mass was maximized with 25-30% of male trees in the orchards (Fig 4D). The results of the seed weight model did not change when we excluded the study site with the highest number of hermaphrodite plants and the study site with the highest percentage of males (S7C Table in Supporting information).

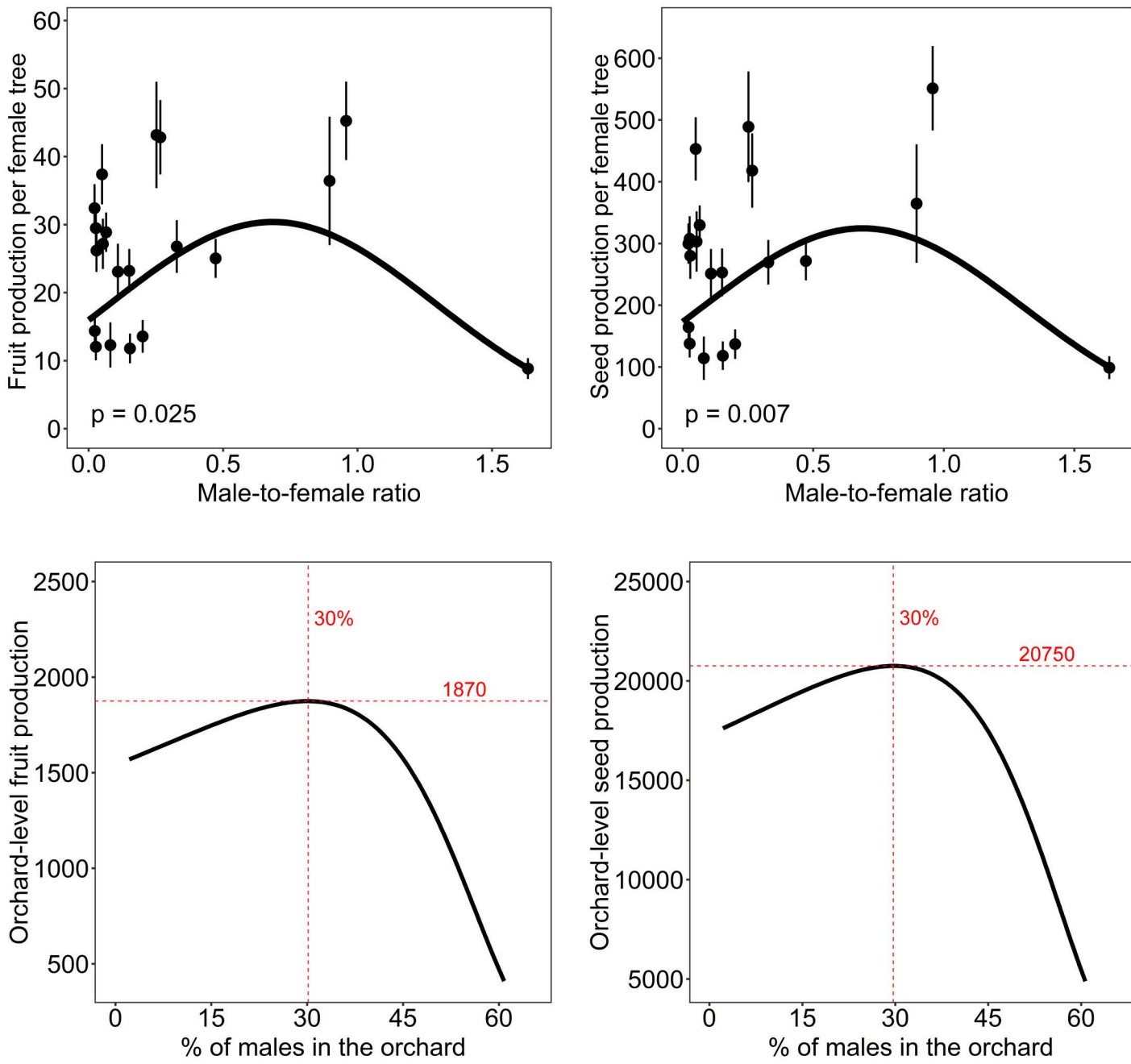

**Fig 3. Effects of local management of orchards on the production of fruits and seeds.** Panels (A) and (B) show the relationships between the male-to-female ratio and fruit and seed production per female tree, respectively. In these panels, lines represent the estimates of the averaged best models, the dots represent average fruit or seed production per year and study orchard, and the vertical lines the standard errors. Panels (C) and (D) show the relationship between the percentage of male trees (the number of male trees per 100 trees) in an orchard and the estimated orchard-level fruit and seed production, respectively (i.e., for all the females in an orchard of 100 trees). Orchard-level fruit production (or seed production) was calculated as the number of fruits (or seeds) produced by 1000 flowers in each tree multiplied by the percentage of females (the number of female trees in an orchard of 100 trees).

## Discussion

Here we showed that both habitat loss and the local management (male-to-female ratio and farming system) influenced the pollinator communities of carob tree orchards. In turn, such changes

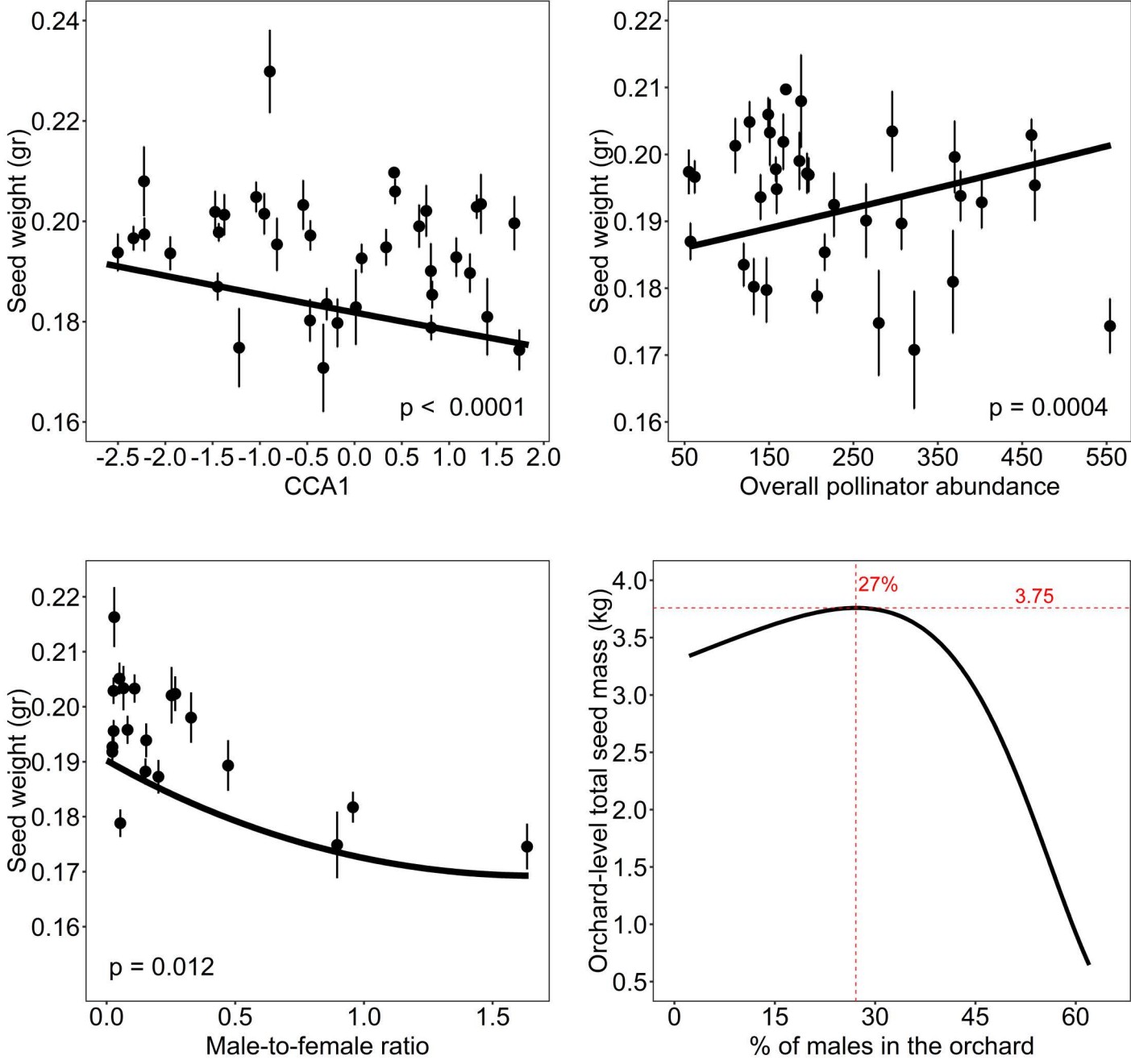

**Fig 4. Pollinator community and local management on seed weight.** Relationships between seed weight and: (A) the first axis of CCA (CCA1); (B) overall pollinator abundance; and (C) male-to-female ratio. Lines in panels (A–C) represent the estimates of the averaged best models, the dots represent the average seed weight per year and study orchard, and the vertical lines the standard errors. Positive values of first axis of CCA (CCA1) represent pollinator community composition related to habitat loss (mainly honeybees), while negative values represent pollinator community composition related to conserved landscape (mainly wild bees and butterflies). (D) Relationship between orchard-level total seed mass and the percentage of male trees (the number of male trees per 100 trees) in an orchard. Orchard-level total seed mass was calculated as the total weight of the seeds produced by 1000 flowers in each tree multiplied by the percentage of females (the number of female trees in an orchard of 100 trees).

in pollinator communities directly affected carob yields, so that carob production was maximized in orchards located within natural landscapes and designed with a 25-30% of male trees.

## Pollinator communities in carob tree orchards

Honeybees were the most frequent visitor in both years, however, we found around 70% more honeybees in 2020 than in 2019. Large differences between years in the abundance of honeybees were also reported by [55], where honeybees went from being the most abundant visitor one year to be almost absent two years later. This high variation in honeybee abundance might be a result of their management, as the installation and removal of hives that hold several tens of thousands of individuals might deeply modify their abundance in crop fields. Wasps and flies (mainly muscoids and small acalyptrate) followed honeybees as the second and third most abundant groups both years, agreeing with [62] and [76] which found wasps and Diptera as the main visitor groups together with honeybees in orchards of Spain and Israel, respectively. Only [76] identified most of the visitors to the genus or species level, naming three bee species, two wasps, one hoverfly, and two flies. Although we only captured a small fraction of all the pollinators we recorded (just 115 out of the 3773 non-honeybee visitors), we were able to identify, by observation or in the laboratory, at least 45 species of wild insects (including at least five of the species identified by [76]) which makes this study the most extensive census of pollinators of carob trees until date.

Most wild pollinators were linked to conserved landscapes, while honeybees were favored in more disturbed areas. These conserved landscapes offer diverse nesting sites and food resources for wild pollinators [77–79]. Pollinators with small foraging ranges like solitary bees and ants require food sources near their nests [80,81], and even long-distance fliers typically forage close to home [61,82]. Hence, natural habitats near the study orchards likely benefited the presence of wild pollinators [83,84]. On the other hand, honeybees showed affinity for disturbed habitats, which might be due to human location of beehives in crop fields [85]. The male-to-female ratio also influenced the composition of pollinator communities. We found a higher number of hoverflies linked to orchards with a higher male-to-female ratio, while the abundance of wasps increased with a lower male-to-female ratio. This was expected because hoverflies feed on both pollen and nectar [86,87], while wasps as *Vespula germanica* (the most common wasp found in our orchards) visit flowers for nectar [88].

Apart from the effect of changes in community composition, we found an overall higher abundance of pollinators in orchards with higher male-to-female ratios. This agrees with previous studies that showed male flowers to be more attractive than female flowers [55,76,89]. Likely this is because male trees produce a much larger number of inflorescences than female trees, have a more intense scent and offer two rewards: nectar and pollen [76,90]. Although we found a slightly higher overall pollinator abundance in fields subjected to ecological farming, the models indicated no significant effect of this variable and no interaction between habitat loss and farming system. Different studies found positive effects of ecological farming on pollinator abundance and diversity [23,24,26], and larger positive effects of ecological farming in more disturbed landscapes [28,29]. In our study, the difficulty of finding ecological orchards that met our landscape needs generated an unbalanced number of conventional vs ecological orchards that might have affected the detection of significant relationships. Future work with a larger number of ecological fields along landscape gradients may help test further these relationships.

## Carob tree production

We recorded a relatively low fruit set compared with the high number of flowers that carob trees produce, which is not rare as the surplus of flowers may be just a mechanism to attract pollinators [91,92]. In addition, fruit production per female tree was lower in 2020 than 2019,

which could be due to biennial bearing [41,93,94], but also to the lower abundance of wild pollinators compared to honeybees in 2020 than in 2019 (54% of the visits vs, 31%, respectively), as wild insects are considered to be more effective than honeybees for crop pollination [9–11]. Contrary to the low fruit set, the seed set was in general relatively high, a pattern also found by other authors [53,62], who suggested that carob trees selectively abort flowers with a smaller pollen load.

We expected an increase in fruit and seed production per female tree related to natural habitats [13,16,18,19], and in orchards under ecological farming [24–27]. However, we did not find direct effects of these two variables on the number of fruits and seeds produced per female tree. We also expected a hump-shaped relationship between the proportion of male trees and fruit and seed production related to density-dependent competitive processes between sexes [37,69]. As expected, local male-to-female ratio influenced the number of fruits and seeds produced per female tree following a hump-shaped relationship, where the yield per female was maximized when the orchards had around 0.6-0.7 males/female tree, whereas the total production at the orchard level was maximized with 25-30% of males from the total. We must note, though, that this hump-shaped relationship was mainly driven by a study orchard with a male-to-female ratio higher than 1.5. The quadratic relationship was lost when this study orchard and the one with mostly hermaphrodite trees (i.e., the two more male biased sites) were removed from the models, while all the other results did not change (S7 Table). Although sex ratio of wild carob trees in natural communities is approximately 1:1 [52,95], which is the common sex ratio of diecious trees [52,96], males were very scarce in most of our study orchards. This unbalanced sex ratio in crop fields is due to extended grafting practices that favor females for production, and prevented us from finding more orchards with ratios higher than 1 to corroborate the hump-shaped relationship between male-to-female ratio and carob production. In any case, this hump-shaped relationship can be easily explained by the classic density-dependent model proposed by Rathcke [37], so that males might facilitate females when at low densities whereas compete with them when at high densities.

However, from the production point of view, the weight of seeds might be at least as important as the number of fruits and seeds produced. Indeed, as many of the most valuable products derived from carob come from the seed [44,45], the production of heavy seeds is essential for producers to increase their profits [43]. We expected that seed weight would be positively related to natural habitat in the landscape and local practices that enhance pollinator diversity and abundance [4,10,13,14]. Here, we showed that the weight of carob seeds increased with overall pollinator abundance, but also with compositions of pollinator communities related to conserved natural habitats (higher abundance of wild bees and butterflies). However, pollinator communities associated with more disturbed landscapes (with more abundance of honeybees and less abundance of wild pollinators) negatively influenced seed weight. The benefits of surrounding natural habitat for agricultural production have been described in several crops (e.g., [97–99]), usually through increased visits of wild pollinators that enhance crop yields. Here we found that conserving natural landscapes has clear benefits on carob seed weight by shaping more effective wild pollinator communities. These results agree with other studies also showing a positive effect of wild pollinator visits on seed weight (e.g., in soybean by [18], in soil rape by [100], or in sunflowers by [101]). Indeed, wild insects are described as more effective pollinators than honeybees for several crop species, as in honeydew melon in Israel [9], alfalfa in South Africa [102] or in several other species around the world as, for instance, almond, buckwheat, cotton, cherry, coffee, and kiwifruit (reviewed in [10]).

Seed weight was not only influenced by the pollinator community, but also by the male-to-female ratio of trees in the study orchards. While the number of carob seeds produced increased as the male-to-female ratio increased, the weight of the seeds decreased. This

opposite effect of the sex ratio on the quantity and quality of seeds might be explained by the well-known trade-off between seed number and size [103,104], that arise from the limited amount of plant resources to invest in reproduction. The intra-specific trade-off between number and size (weight) has been described for multiple species (e.g., [105–108]). In crops, this trade-off is especially known in fruit trees, such as kiwifruit [109], apple [110], or peach [111], and is one of the reasons for pruning branches, as the reduction in the number of potential fruits increases the size of the remaining ones. In carob trees, the production of heavier seeds is of interest to farmers to increase their economic profits, as currently seeds are the most profitable part of carob fruits (17-27€/kg in 2022) [43]. Here, we estimated that orchard-level maximum total yield in terms of weight occurs with 25-30% of males in the field. Therefore, implementing strategies for natural habitat and wild pollinator conservation and maintaining adequate male-to-female ratios will translate into a higher market value and increasing farmer's profits [15,100].

## Conclusions

Here we showed that natural landscapes promote wild pollinator communities (mostly wild bees) in carob tree orchards, while disturbed landscapes held primarily managed honey-bees. The carob tree male-to-female ratio also influenced the abundance of pollinators at the orchard level, and modulated the production of fruits and seeds. Wild pollinator communities associated with conserved landscapes increased the weight of seeds, which are the most profit-able part of carob production. To optimize yields and profits, we suggest an orchard composition with 25-30% male trees, closely surrounded by natural habitats.

## Supporting information

**S1 Fig. Pictures of carob tree flowers and orchard.**
(PNG)

**S2 Fig. Map of the 20 study orchards.**
(PNG)

**S1 Table. Landscape and local characteristics of the study orchards.**
(XLSX)

**S2 Table. Results of Moran's I tests to evaluate spatial autocorrelation in the response variables.**
(XLSX)

**S3 Table. Pollinator visits by group in the 20 study orchards for each sampling year.**
(XLSX)

**S4 Table. Proportion of visits to female and male carob flowers for each of the pollinator functional groups.**
(XLSX)

**S5 Table. List of captured wild pollinator species.**
(XLSX)

**S6 Table. Results of canonical correspondence analysis to investigate pollinator community composition; re-analysis excluding the study orchard with highest male-to-female ratio and the one with mostly hermaphrodite trees.**
(XLSX)

**S7 Table. Results of model-averaging from model selection of GLMMs; re-analysis excluding the study orchard with the highest male-to-female ratio and the one with mostly hermaphrodite trees.**
(XLSX)

## Acknowledgments

We are very grateful to the owners of the study orchards who allowed us to work on their properties. We also thank Annika Salzberg and two anonymous reviews for their critical and constructive comments on an earlier version of this manuscript.

## Author contributions

**Conceptualization:** Amparo Lázaro.

**Data curation:** Carmelo Gómez-Martínez, Miguel A. González-Estévez, Peter Unglaub.

**Formal analysis:** Carmelo Gómez-Martínez, Amparo Lázaro.

**Funding acquisition:** Amparo Lázaro.

**Investigation:** Carmelo Gómez-Martínez, Miguel A. González-Estévez, Indradatta deCastro-Arrazola, Amparo Lázaro.

**Methodology:** Carmelo Gómez-Martínez, Amparo Lázaro.

**Project administration:** Amparo Lázaro.

**Resources:** Miguel A. González-Estévez.

**Supervision:** Amparo Lázaro.

**Writing – original draft:** Carmelo Gómez-Martínez, Amparo Lázaro.

**Writing – review & editing:** Carmelo Gómez-Martínez, Miguel A. González-Estévez, Indradatta deCastro-Arrazola, Peter Unglaub, Amparo Lázaro.

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
