## [Decision Letter · Decision Letter 0]

1 Sep 2024

PONE-D-24-27278Landscape conservation and orchard management influence carob tree yield through changes in pollinator communitiesPLOS ONE

Dear Dr. Lázaro,

Thank you for submitting your manuscript to PLOS ONE. After careful consideration, we feel that it has merit but does not fully meet PLOS ONE’s publication criteria as it currently stands. Therefore, we invite you to submit a revised version of the manuscript that addresses the points raised during the review process.

Your manuscript has been revised by three reviewers that provide feedback to improve its clarity and quality. Reviewers highlight the importance and relevance of the study, but they also identify some issues which require attention. I am specially concerned about the possible impact of outliers and spatial autocorrelation on your results, please pay special attention to these issues which have been also highlighted by the reviewers. Overall, these issues should be feasible to address through caraful revision. 

We look forward to receiving your revised manuscript.

Kind regards,

Vicente Martínez López

Academic Editor

PLOS ONE

Journal Requirements:

2. In your Methods section, please provide additional information regarding the permits you obtained for the work. Please ensure you have included the full name of the authority that approved the field site access and, if no permits were required, a brief statement explaining why

"We are very grateful to the owners of the study orchards who allowed us working on their properties. This study was supported by the project CGL2017-89254-R, financed by the Spanish Ministry of Science and Innovation, FEDER Funds and the Spanish State Research Agency, and by the project PRPPID2020-117863RB-I00, financed by the Spanish Ministry of Science and Innovation and the Spanish State Research Agency (MCIN/AEI/10.13039/501100011033). CGM was supported by a FPI predoctoral contract financed by the Spanish Ministry of Economy and Competitiveness, the Spanish Research Agency, and European Social Funds (FPI PRE2018-083185, Call 2018). IMEDEA is an accredited “Maria de Maeztu Excellence Unit” (Grant CEX2021-001198, funded by MCIN/AEI/10.13039/501100011033) (period 2023-2027)."

"This study was supported by the project CGL2017-89254-R, financed by the Spanish Ministry of Science and Innovation, FEDER Funds and the Spanish State Research Agency, and by the project PRPPID2020-117863RB-I00, financed by the Spanish Ministry of Science and Innovation and the Spanish State Research Agency (MCIN/AEI/10.13039/501100011033). CGM was supported by a FPI predoctoral contract financed by the Spanish Ministry of Economy and Competitiveness, the Spanish Research Agency, and European Social Funds (FPI PRE2018-083185, Call 2018)"

"This study was supported by the project CGL2017-89254-R, financed by the Spanish Ministry of Science and Innovation, FEDER Funds and the Spanish State Research Agency, and by the project PRPPID2020-117863RB-I00, financed by the Spanish Ministry of Science and Innovation and the Spanish State Research Agency (MCIN/AEI/10.13039/501100011033). CGM was supported by a FPI predoctoral contract financed by the Spanish Ministry of Economy and Competitiveness, the Spanish Research Agency, and European Social Funds (FPI PRE2018-083185, Call 2018)"

5. Please include captions for your Supporting Information files at the end of your manuscript, and update any in-text citations to match accordingly. Please see our Supporting Information guidelines for more information: http://journals.plos.org/plosone/s/supporting-information .

Reviewers' comments:

Reviewer's Responses to Questions

**Comments to the Author**

1. Is the manuscript technically sound, and do the data support the conclusions?

Reviewer #1: Yes

Reviewer #2: Partly

Reviewer #3: Yes

2. Has the statistical analysis been performed appropriately and rigorously? 

Reviewer #1: Yes

Reviewer #2: Yes

Reviewer #3: Yes

3. Have the authors made all data underlying the findings in their manuscript fully available?

Reviewer #1: Yes

Reviewer #2: Yes

Reviewer #3: Yes

4. Is the manuscript presented in an intelligible fashion and written in standard English?

Reviewer #1: Yes

Reviewer #2: Yes

Reviewer #3: Yes

5. Review Comments to the Author

Reviewer #1: This current version of this study demonstrated that the % natural habitat, overall pollinator abundance and male-to-female ratio significantly affected the seed production of carob trees. Basically, the experimental method is sound, the result is interesting and the discussion is reasonable. However, there are still some issues that need to be revised and adjusted the prior to publication.

1. In the abstract, please clarify the current problems that need to be solved in the production of carob, that is, the purpose of the study.

2. There is a lack of review or comparison of similar studies in the preface, such as: the impact of different landscape types or orchard management on yield? The research value and significance of long-horned beans are not clear.

3. L98～L120: The introduction of research crops is suggested to be included in the preface to comprehensively review the research value and significance of carob。

4. Supplement orchard information (variety, planting area, tree age, flowering, fruiting, climate, elevation, etc.) in material method。

5. L19～L20，L124: Orchard local management (farming system: conventional vs. ecological)’ and ‘Fig. 1’ mention ‘conventional vs. ecological,’ but the specific concepts or definitions related to this are found in lines 153–156. It is suggested to unify and organize the research orchard’s location information, management information, and so on, to enhance the readability of the paper.

6. L129～L130:“it cannot be totally discarded that they include different cultivars，as this information was unknown by the farmers.”. Although different orchards have different varieties, it should be explained here whether the different varieties affect the assessment and comparison of yields between different orchards.

7. L159～L161:“The male-to-female ratio in carob orchards of Mallorca is generally low, as farmers obtain female carob trees by grafting female branches in male rootstocks to increase the number of producer trees”. It is recommended to include this content in the introduction or discussion section for elaboration

8. It is recommended that the author include a diagram illustrating pollinator visits to flowers. This addition will enhance the content of the paper and provide readers with a clearer understanding of carob as an economic crop.

9. L162～L163: “except for one study orchard where most of the individuals (95.7%) were hermaphrodite”. When evaluating the impact of the male-female ratio on yield, should the data from this orchard be excluded? The results will be more accurate and will be more instructive for future carob planting.

10. L195-197：I have a concern: is it sufficient for each tree to randomly select a branch to evaluate yield metrics?

11. L291～L292: According to the result description part “Fruit production per female tree varied between 8.8 and 45.2 fruits in one thousand flowers”, is it more appropriate to change “Fruit production per female tree” to “Number of fruits per female tree”? Seed yield is the same as above. L295 “seed weight” is changed to “seed production”. Like lines L446~L447 (the number of carob seeds, the weight of the seeds), this is easier to understand.

12. Judging from the results in Table 1 A, there is a positive correlation between overall pollinator abundance and farm system, but the relationship is not significant (P=0.097), rather than “the effect was only marginally significant” as the author said.

13. L433～L435:“On the contrary, pollinator communities associated with more disturbed landscapes (high abundance of honeybees) negatively influenced seed weight.”. Does higher bee abundance have a negative effect on seed weight? This conclusion seems inconsistent with common sense? Although wild bees are more efficient pollinators, bees are also efficient pollinators of flowering plants.

Reviewer #2: The present study aims at understanding how carob tree yield are influenced by surrounding natural areas and orchard management, from the pollination perspective. Being a dioecious crop, the authors also explore the impact of male/female ratio and assess the optimal ratio that provides higher values of pollinator visitation and production. Results show that orchards with higher amount of surrounding natural areas received higher wild pollinator visitation, but lower visitation by honeybees. Production did not observe the same patterns but was positively related with pollinator visitation in conventional orchards. Results also show that production is maximized when 20/30% males occur in the orchard, which is an important results in the management perspective.

The crop being studied gives relevance to the study, being an economic important crop. It is also an understudied crop regarding pollination and represents an interesting reproductive system (dioecy), which means additional difficulties regarding pollination, as male/female ratios aspects, as explored here.

The introduction is focused on the explored topics and the aims are clearly given. I only have small suggestions (given below after the major remarks). However, I suggest some changes in the Methods and Results section, to increase clarity and facilitate results observation for the reader. I also think there is some work needed on the Abstract. I provide them below:

- I suggest improving the abstract and reworking some parts of it. Specifically, I suggest improving how your results impact management in this crop. I got a bit confused with the ecological vs. conventional observed results (after reading the full article I understand better, but it becomes visible that the abstract needs to be improved on this part). I would not have the “ecological farming tended to increase (…)” phrase, once the results were marginally significant and the most interesting part is on the conventional orchards, where higher abundance of wild pollinators was observed.

- “Study crop”, methods section, has some duplicated information with the Intro, so I suggest revision to remove them, or even, remove the subsection (and put essential information to the introduction). E.g. of duplicates - Spain being the main producer (with 60-80 thousand tons per year produced). There is also some misplacement – some of the information given in the methods would be great to have before the aims are given, as the nice explanation that both pods and seeds are used (which is also given in the intro but is less complete) or the problems with the female spatial isolation, which hinders pollination. Regarding the crop itself, I missed specific information on pollinator dependence of the crop, there is any information/levels already published on this more than the “high level”? It can be important for the reader to assess the ratio pollinator importance vs. non-animal pollination.

- Landscape characterization: Can you provide more information on this? The authors state that they used natural area, but before they mentioned natural and semi-natural. Did you use both or only natural areas? If you used only natural areas, they need to be referred specifically. If you used both I suggest using a different name that includes both (semi-natural for ex.). Also, why is water bodies under crops, can you provide more context to it? I also do not completely understand the pastures (I am associating them with human management) under natural/semi-natural areas, why they are under this group of semi-natural habitat?

- Can you provide, at least as supplementary, graphical representations of the results regarding overall pollination and its relationship with the natural area amount and field management? It is very hard to follow and understand the results only with the table and the CCA results, and graphs as in Figure 2B would improve reader comprehension and give a better idea of the results, even when no differences were detected (or they were marginal).

Overall, I miss some information about the orchard management and flowering related aspects, that may be important to form conclusions and put results in perspective:

- A common problem in dioecious crops are the mismatches between female and male flowering. Do they exist in carob? Or is this not a problem at all? Did you account or assessed for this? And was the flower density the same between orchards?

- The manuscript is missing information regarding distance between trees / tree density? Do they changed between orchards? And what is the distance/density often practiced, can you provide this information?

- Once honeybees were the major pollinator in both years, I am considering that hives occur in the landscape or are applied in the orchards. Can you provide more information on this? Do you have information of presence of hives on the different orchards?

Minor remarks/questions:

Why did you selected the 1km radius?

Did you notice any differences in pollinator visitation composition between male and female parts? Does the female produces any reward (if yes, I suggest giving this information in the manuscript).

Please revise the number of decimals through the manuscript. There are different decimal numbers being used.

The hump-shaped relationship – as you stated in discussion, the relationship is highly defined by one orchard. The statement in the abstract should follow the same consideration, and not be stated as a strong observed result.

Line 44-45 – The article you cite indicates 75% of cultivated plants depend on insect pollination. 35% is directed to global production. Please, revise the phrase.

Line 55 – what do you mean with “ecological crops”? Please revise the phrase.

Line 61 – Dioecious crops rely on both sexes for sexual reproduction, but asexual can still occur. I suggest adding the sexual to the phrase.

Line 70 – I suggest explaining what is a polygamotrioecious species.

Line 80 – do you have another word for palliate (moderate/mitigate?)? The same for line 195, with haphazardly (randomly?)? they are non-common words, so I suggest using more known words to facilitate reading to non-native English readers.

Line 83-84 – I would use gradient of percentage of natural area instead of the opposite, once it is what is being used through the manuscript.

Line 105 – I suggest using -main- instead of -first-.

Line 112-113 – In my opinion, information like this may be unnecessary, once it is about the natural occurrence of the species. It suggest removing it.

Line 153 – I suggest – the remaining 13 were under conventional management.

Line 161 – When you indicate hermaphrodite trees, are you referring to these grafted female/male trees or originally hermaphrodite trees? Please, revise accordingly.

Line 196 – what do you mean with inflorescence buds? Do you mean closed/early stage inflorescences? I suggest using the open/closed for both inflorescences and flowers.

Line 202 – I suggest taking out the “in them”. Maybe per branch work better.

Line 266 – there is a character not recognized.

Line 285-288 – it is confusing that numbers per year are presented for total and honeybees but then you stop giving them per year. Can you provide per year for all? Wild pollinators were higher contributors in 2019, but this way we cannot identify which wild groups were responsible for it.

Line 345-347 – this line is very confusing. I do not understand the message you want to give.

Lines 416-418 – The ratio found in nature and that dioecious usually occur in a 1:1 ratio seems unnecessary information, I suggest removing it. Males are scarce due to a producer decision, what happens in nature does not bring much to the discussion. Being females the trees that produce it is normal that males occur in a scarcer number, and you data shows that (the optimal levels is 20/30% of males).

Also, it is only a suggestion/preference, but it would be very nice to have some photographs of the male and female inflorescences/flowers, as well as of the usual orchards (it can be as a supplementary). The crop is very different from the usual animal pollinated crops, so it would be nice this to establish how results relate to flower structure, for example.

Reviewer #3: General comments:

I find this paper to be a valuable contribution to the literature on local and landscape level effects on insect ecosystem service provisioning. However, there are several larger edits and many small edits the authors should make before publication.

As it stands, the introduction section hits the major points but lacks depth. For example, on lines 56-57 the authors predict that “the negative effects of non-ecological management on pollinator abundance and richness may be stronger in simplified than complex landscapes” but do not elaborate on why this might be the case. A few lines later (64-66) they reference “density-dependent processes of facilitation-competition between sexes.” While I was able to later figure out what this meant through context of the study, it would be beneficial to readers’ understanding to explain this further in the introduction.

A general question I had throughout reading this paper is whether the authors were aware whether the farms they worked at were using supplementary honey bee hives. Honey bees were discussed at length throughout the paper, but there was no information included about how common adding honey bee hives to farms/orchards is on Mallorca Island or whether there is already a population of wild honey bees there. This information is crucial to this study, where they found more than half the visits to carob trees were by honey bees. See below further questions about this:

Lines 45-46: “Although managed honeybees are commonly used to pollinate crops, wild insects are often more effective pollinators” Are they commonly used on Mallorca?

(lines 311-313) “whereas more disturbed landscapes (low percentage of natural habitats) were related to an increase in the abundance of managed honeybees.” And (lines 366-367) “Most wild pollinators were linked to conserved landscapes, while honeybees were favored in more disturbed areas.” But was this due to their preference or was this just where farmers were placing hives in agricultural sites?

(lines 353-356) “This high variation in honeybee abundance might result of their management, as the installation and removal of hives that hold several tens of thousands of individuals [69] might deeply modify their abundance in crop fields.” – this citation references a New York State beekeeping guide, but do beekeeping practices on a small island in the Mediterranean match those of the continental USA?

Lines 372-373: “honeybees showed affinity for disturbed habitats, which might be due to the location of beehives in crop fields” Is this showing an affinity, or simply showing where humans placed them?

Line edits:

Abstract

Lines 18-19: Should be “on Mallorca Island” (on instead of in, capitalize Island)

Line 23: “surrounded by larger natural areas” – to be very picky about your wording here, this could imply that the natural areas are less fragmented, but that’s not your intended meaning. I would reword this to say “surrounded by a greater proportion of natural landcover” or something along those lines, to be clear you’re talking about proportion land cover and not another landscape characteristic like connectivity or fragment size.

Introduction

Line 39: “poses” instead of “supposes” is the correct word here.

Line 41: “declines” instead of “decline”

Line 44: remove the word “the” in “because the 35% of cultivated plants”

Line 50: “e.g.” is not needed here

Line 51: “landscapes” shouldn’t be plural

Line 52: Use “less” instead of “little”

Lines 52-53: I would also suggest the citation Grab et al. 2019, “Agriculturally dominated landscapes reduce bee phylogenetic diversity and pollination services”

Line 53: Remove “besides” and add “also” – “Local crop management may also have additional effects on [...]”

Line 55: You say “ecological” crops, but all four of your citations look at organic crops (with one exception also looking at flowering strips) and as far as I can see don’t seem to mention ecological farming. I may have missed something in the papers, but these citations mostly don’t support the statement unless you change “ecological” to “organic”.

Line 59: change to “depend on the crop breeding”

Line 60: You use “particularly” twice in this sentence, remove the first instance. (So; “Dioecious crops, which rely on [...], might be particularly vulnerable [...]”

Line 62: Clarify “most efficient one.” I suggest changing to “as well as to the loss of the most efficient species”

Line 63: “crop” (not crops)

Line 64: Missing a word, should be “And equilibrated sex ratio”

Line 69: The word “extended” doesn’t make sense here, do you mean it grows extensively throughout the Mediterranean?

Line 73: “thousand tons produced per year” (add “produced”)

Line 74: “uprising” is incorrect, should be “rising”

Line 80: “Palliate” is out of place, usually used in the context of disease. I suggest “alleviate” instead.

Line 84: “On Mallorca Island” instead of “in”

Lines 89-92: needed small grammar edits (in italics): “We expected that orchards within conserved landscapes and subjected to ecological farming would hold a higher pollinator abundance and have community compositions that enhanced carob production; and that the negative effect of habitat loss on crop production would be less pronounced in ecological farms.”

Lines 90-91: What community composition specifically would enhance carob production? Would that be increased species diversity in general, or increased presence of certain species you know to be good at pollinating carob specifically?

Lines 92-93: Why did you expect male trees to be more appealing to pollinators? I’d like more clarification in the introduction about what you would expect the ideal sex ratio in carob orchards to be and citations to back up your prediction here.

Materials and Methods

Lines 101-104: I would split this sentence into two, like this: “The carob pods, seeds, and gum have many nutritional, biochemical, and clinical applications. These include livestock feeding, food and beverages, bioenergy production, in the pharmaceutical industry in pomades, anti-celiac ingredients, pills, and dental paste.” (I question here though whether pomade should be grouped in with pharmaceutical products? Is a hair product a pharmaceutical? This is entirely up to the authors to change this or not.)

Line 105: “Largest” producer instead of first producer is more grammatically correct

Line 106: “concentrate” isn’t quite the right word, I would use “contribute” instead

Line 107: Rewording necessary for grammar, my suggestion: “Carob market prices are cyclic, with long gaps and short high peaks that have historically happened in 1984, 1994, 2006, 2011 [39], and recently in 2022, when the prices of Spanish and Portuguese production in the international market raised from [...]” However, I’m not sure it’s necessary to put this much detail about which years specifically. The authors might consider saying instead that market prices tend to spike every 10 years on average, and then fall again. Similarly, at the end of this sentence instead of saying “raised from 0.8€/kg to more than 2€/kg for whole pods and from 7€/kg to 27€/kg for seeds” it may make your point clearer to say something along the lines of “Recently in 2022, the price of whole pods (per kilogram) more than doubled and the price of seeds nearly quadrupled in the international market.” The exact price differences aren’t relevant to your work, but the overall economic impact is.

Line 118: I assume male tree abundance in orchards is low because farmers selectively plant females? Be clear in the text that this is an artificially skewed ratio.

Line 123: How large were each of the orchards? I would recommend including mean +/- SE of orchard size. Were they statistically similar sizes, and if not, was that taken into account in your analyses? This could significantly impact your results if the orchard sizes varied widely.

Line 125: Replace “see below” with a reference to a specific section, this makes it unclear to the reader whether you mean to reference the next section of text or a figure lower down.

Line 126: “from the center of the orchards” was missing the word “of”

Lines 126 and 136: You selected sites at minimum 1.5km from each other, but measured 1km buffers, leaving the option for the edges of sites to be only 0.5km apart. Did you test for spatial autocorrelation of sites? Looking at your map in the supporting information, some sites appear very close together and I would recommend running a Mantel test to check.

Line 155: “does not use” instead of “do not use”

Lines 162-163: This seems like a very extreme outlier in terms of sex ratio, I’m curious why this site was still included in the study when it seemingly has so much potential to skew results.

Lines 171-173: How many observers were there? You say “we,” how many people were observing at a time?

Lines 181-183: I would call these general taxonomic groups, not functional. Calling them functional groups implies each group provides a unique service, but you are assuming these groups are all performing the same service, pollination.

Line 251: “Farming” is misspelled

Lines 253-255: If I’m not mistaken, the results of this GLMM is in Table 1a – however, I don’t see the interaction reported there or in the discussion section. If these results weren’t significant, you should still report them – and if the interaction was taken out of the model you should note that as well.

Line 272: Should be “fulfill” instead of “fulfilled”

Line 365: “extent” should be “extensive”

Line 411: “being the yield per female maximized” should be “where the yield per female was maximized”

Lines 413-416: “We must note, though, that this hump-shaped relationship was mainly driven by a study orchard with a male-to-female ratio higher than 1.5, and that the quadratic shape of this relationship was lost if that study orchard was not included in the analyses.” Do you think this outlier site, which you know affected the results of the referenced analysis, could also have affected your other results? Did you re-run your other analyses as well with this site removed?

Line 425: Typo, “as least as important” should be “at least as important”

Line 256: “Nowadays” is too informal for use here, use “currently” or “at the time this study was conducted”

Line 457: The exact prices of carob aren’t necessary to include, though if the authors find it important I would include just the 2022 price.

Line 469: Should be “associated with” instead of “associated to”

Line 470: “For that [...]” it’s unclear what “that” is that you refer to. I would suggest changing to “To optimize yields and profits, we suggest an orchard composition with 20-30% male trees, closely surrounded by natural habitats and subjected to ecological practices.”

References

Throughout, I found inconsistent use of italics for species names. I know this is a common problem with reference software, where exporting a bibliography removes text formatting, so please go through manually and ensure that all genus and species names are italicized.

There is inconsistency in what punctuation the authors have used after the year of publication, and between the journal name and year. I can see “.” , “;” and “:” all used in different citations after the year. (ex. Citation 84 has “1996.” While citation 85 has “1994;”. Please standardize to one form of punctuation between elements across all citations.

Additionally, there are multiple occurrences where spaces are missing or capitalization is incorrect. I have tried to find all instances of these below, but please go through and check all references for these small errors as well. Also, please standardize the citations so they are all left-aligned. Some are center-justified, which makes the spacing inconsistent.

Line 618: “Research1997” space needed

Line 628: “Madrid.2021.” space needed

Line 652: “Rural2023” space needed

Line 660: “populationof” space needed

Line 673: “Chapman” and “Hall” should both be capitalized

Line 678: “springer” should be capitalized

Line 683: “Cambridge University Press” – each word should be capitalized

Line 731: “1986” is written twice

Line 733: I assume “Madrid” should be capitalized?

Line 743: There should be a period after “editor” to clearly separate the name of the author from the title.

Lines 777 and 780: “Phytologist” should be capitalized

Line 788: Space needed between “gala” and “apple”

6. PLOS authors have the option to publish the peer review history of their article (what does this mean? ). If published, this will include your full peer review and any attached files.

**Do you want your identity to be public for this peer review?** For information about this choice, including consent withdrawal, please see our Privacy Policy .

Reviewer #1: No

Reviewer #2: No

Reviewer #3: **Yes: ** Annika Salzberg

---

## [Author Response · Author response to Decision Letter 0]

16 Oct 2024

Authors' Responses to Questions

Reviewer #1: This current version of this study demonstrated that the % natural habitat, overall pollinator abundance and male-to-female ratio significantly affected the seed production of carob trees. Basically, the experimental method is sound, the result is interesting and the discussion is reasonable. However, there are still some issues that need to be revised and adjusted the prior to publication.

REPLY: Thank you very much for your valuable comments and your positive assessment. We have revised the manuscript according to your suggestions. Please find below our responses to your comments.

1. In the abstract, please clarify the current problems that need to be solved in the production of carob, that is, the purpose of the study.

REPLY: We have rewritten several parts of the abstract to clarify this (lines 18-26).

2. There is a lack of review or comparison of similar studies in the preface, such as: the impact of different landscape types or orchard management on yield? The research value and significance of long-horned beans are not clear.

REPLY: In this version, we have included a larger comparison with more studies in the introduction (lines 54-66), but focusing always in the variables that we include in the models (i.e., loss of natural areas in the landscape, ecological farming, and sex ratios).

3. L98～L120: The introduction of research crops is suggested to be included in the preface to comprehensively review the research value and significance of carob。

REPLY: As indicated, we have included the information of this section in the introduction (lines 80-100), and have removed the ‘study crop’ section to not repeat information.

4. Supplement orchard information (variety, planting area, tree age, flowering, fruiting, climate, elevation, etc.) in material method。

REPLY: We added the information available in Table S1 in the Supplementary Information and in the main text (lines 134-136 and 140-141). Flowering and fruiting are described in the methods section. The new information given included elevation, climate, orchard area, tree density and the use of honeybee hives. Other variables such as variety or tree age could not be added because it is unknown by the farmers.

5. L19～L20，L124: Orchard local management (farming system: conventional vs. ecological)’ and ‘Fig. 1’ mention ‘conventional vs. ecological,’ but the specific concepts or definitions related to this are found in lines 153–156. It is suggested to unify and organize the research orchard’s location information, management information, and so on, to enhance the readability of the paper.

REPLY: We have added a short definition of ecological crops in the introduction as suggested (lines 61-63). As in the abstract we have word limitations and the concept of ecological crops is generally well understood, we have not included this definition in the abstract. We prefer to keep separated the sections for local management, landscape and study orchards as the two formers are the main predictors in our study. However, following your suggestions, we have moved some information about the crop and its management to the introduction to enhance readability (lines 80-100).

6. L129～L130:“it cannot be totally discarded that they include different cultivars，as this information was unknown by the farmers.”. Although different orchards have different varieties, it should be explained here whether the different varieties affect the assessment and comparison of yields between different orchards.

REPLY: We agree this would be ideal, but farmers in Mallorca Island do not know the cultivars they have, so this information is not available.

7. L159～L161:“The male-to-female ratio in carob orchards of Mallorca is generally low, as farmers obtain female carob trees by grafting female branches in male rootstocks to increase the number of producer trees”. It is recommended to include this content in the introduction or discussion section for elaboration

REPLY: This is now included in the introduction (lines 97-100). In the discussion section it was already included (see lines 519-522).

8. It is recommended that the author include a diagram illustrating pollinator visits to flowers. This addition will enhance the content of the paper and provide readers with a clearer understanding of carob as an economic crop.

REPLY: We added a figure with pictures of flowers and of a pollinator visiting a carob inflorescence in the Supporting Information (Fig S1) and referred to it in the main text (line 91).

9. L162～L163: “except for one study orchard where most of the individuals (95.7%) were hermaphrodite”. When evaluating the impact of the male-female ratio on yield, should the data from this orchard be excluded? The results will be more accurate and will be more instructive for future carob planting.

REPLY: Our species in polygamo-trioecious (with female, male, hermaphroditic plants, and plants having bisexual and male flowers and others having bisexual and female flowers) (lines 87-90). With our analyses we aimed at considering all the variability in sex-ratios displayed by the orchards in Mallorca island, that includes from crops with mainly no males, to crops with very few percentage of females, and hermaphrodite plants which have both sexes at approximately equal rates. This is why we prefer to leave the analyses as we had them in the main text. However, you suggested removing this field with mostly hermaphrodite plants, and another reviewer suggested removing the orchard with more males. Following the suggestions of both of you, we have now included in the Supporting information (Tables S6 and S7) new analyses excluding these two orchards. The results regarding pollinators and seed weight remain consistent, however, the quadratic effect on fruit and seed set is lost when these two orchards are removed, which is normal because these two localities are those with the most male- biased sex in our dataset (mostly the other orchard with a high percentage of male trees). These new results are now mentioned in different parts of the manuscript (Abstract lines 33-34; methods lines 309-312; results lines: 355-357, 376-379, 400-403, 413-415; discussion lines 514-516; Tables S6 and S7)

10. L195-197：I have a concern: is it sufficient for each tree to randomly select a branch to evaluate yield metrics?

REPLY: For the objectives of our work, we needed to accurately control the number of flowers that yielded fruit, to accurately estimate fruit and seed production. Because the number of inflorescences and flowers in carob trees are massive, counting flowers with precision in a wide part of the tree becomes almost impossible, whereas by randomly selecting a branch, we could control flowering and fructification with a high precision. To avoid biases and get good sample sizes, we selected very large branches (37.85 ± 1.44 mean (SE) inflorescences and 969.55 ± 33.91 flowers per marked female tree), which is now clarified in the text (lines 217 and 221), and therefore, we think that our sampling was adequate.

11. L291～L292: According to the result description part “Fruit production per female tree varied between 8.8 and 45.2 fruits in one thousand flowers”, is it more appropriate to change “Fruit production per female tree” to “Number of fruits per female tree”? Seed yield is the same as above. L295 “seed weight” is changed to “seed production”. Like lines L446~L447 (the number of carob seeds, the weight of the seeds), this is easier to understand.

REPLY: We have changed seed and fruit production to ‘number of fruits and seeds’, as suggested (lines 505-506, 509, 527).

12. Judging from the results in Table 1 A, there is a positive correlation between overall pollinator abundance and farm system, but the relationship is not significant (P=0.097), rather than “the effect was only marginally significant” as the author said.

REPLY: We have changed it to say that it was not significant (line 376).

13. L433～L435: “On the contrary, pollinator communities associated with more disturbed landscapes (high abundance of honeybees) negatively influenced seed weight.”. Does higher bee abundance have a negative effect on seed weight? This conclusion seems inconsistent with common sense? Although wild bees are more efficient pollinators, bees are also efficient pollinators of flowering plants.

REPLY: It is not higher bee abundance that have a negative effect on seed weight, it is that crop fields within landscapes where pollinator communities are dominated by honeybees (instead of wild pollinators) have lower seed weight. We have slightly modified the sentence to clarify (lines 535-536).

Reviewer #2: The present study aims at understanding how carob tree yield are influenced by surrounding natural areas and orchard management, from the pollination perspective. Being a dioecious crop, the authors also explore the impact of male/female ratio and assess the optimal ratio that provides higher values of pollinator visitation and production. Results show that orchards with higher amount of surrounding natural areas received higher wild pollinator visitation, but lower visitation by honeybees. Production did not observe the same patterns but was positively related with pollinator visitation in conventional orchards. Results also show that production is maximized when 20/30% males occur in the orchard, which is an important results in the management perspective.

The crop being studied gives relevance to the study, being an economic important crop. It is also an understudied crop regarding pollination and represents an interesting reproductive system (dioecy), which means additional difficulties regarding pollination, as male/female ratios aspects, as explored here. The introduction is focused on the explored topics and the aims are clearly given. I only have small suggestions (given below after the major remarks). However, I suggest some changes in the Methods and Results section, to increase clarity and facilitate results observation for the reader. I also think there is some work needed on the Abstract. I provide them below:

REPLY: Thank you very much for your positive assessment and your comments and suggestions to improve the manuscript. Following your suggestions, we have modified several parts of the abstract, methods and results to improve clarity for the reader. See below the specific changes in this version.

- I suggest improving the abstract and reworking some parts of it. Specifically, I suggest improving how your results impact management in this crop. I got a bit confused with the ecological vs. conventional observed results (after reading the full article I understand better, but it becomes visible that the abstract needs to be improved on this part). I would not have the “ecological farming tended to increase (…)” phrase, once the results were marginally significant and the most interesting part is on the conventional orchards, where higher abundance of wild pollinators was observed.

REPLY: We have now clarified in the abstract (line 26-29), results (lines 373-376), and discussion (lines 483-485) that there is no significant difference between ecological and conventional farming, despite being overall pollinator abundance is slightly higher in ecological compared to conventional orchards. Consequently, we have removed the comment about ecological farming from the conclusions (line 567). We have also modified the last part about the management of crops giving priority to landscape effects both in the abstract (lines 35-38), in the first part of the discussion (lines 483-485), and in the conclusions (lines 569-572).

- “Study crop”, methods section, has some duplicated information with the Intro, so I suggest revision to remove them, or even, remove the subsection (and put essential information to the introduction). E.g. of duplicates - Spain being the main producer (with 60-80 thousand tons per year produced). There is also some misplacement – some of the information given in the methods would be great to have before the aims are given, as the nice explanation that both pods and seeds are used (which is also given in the intro but is less complete) or the problems with the female spatial isolation, which hinders pollination. Regarding the crop itself, I missed specific information on pollinator dependence of the crop, there is any information/levels already published on this more than the “high level”? It can be important for the reader to assess the ratio pollinator im portance vs. non-animal pollination.

REPLY: This section has now been moved to the introduction as suggested (lines 80-100), and the old section about the study crop has been removed to avoid duplicates. Pollinator dependence of the crop was mentioned before, but now we have extended the sentence to include more details (lines 92-94).

- Landscape characterization: Can you provide more information on this? The authors state that they used natural area, but before they mentioned natural and semi-natural. Did you use both or only natural areas? If you used only natural areas, they need to be referred specifically. If you used both I suggest using a different name that includes both (semi-natural for ex.). Also, why is water bodies under crops, can you provide more context to it? I also do not completely understand the pastures (I am associating them with human management) under natural/semi-natural areas, why they are under this group of semi-natural habitat?

REPLY: We now provide more information as requested. The landscape is not ‘under the crops’, it is within 1-km radius surrounding the crops (lines 156, this is why there can be water bodies. The study orchards had a mean size of 4.41 ± 1.09 (which is now detailed in line 135 in the main text). We now also explain why we selected this buffer radius (lines 156-157). Regarding natural and semi-natural habitats, we have rewritten the text to clarify (lines 159-161). Within natural habitats we have natural habitats (mix of shrub and pasture, mix of wood and pasture, woodland, perennial and coniferous forest, and shrubland) and one single semi-natural habitat (pastures) that correspond to a low percentage of areas. Extensive pastures in the Mediterranean are not natural habitats because they are to some extent managed by human by livestock grazing, but behave very much as natural habitats as they are highly diverse in terms of flowering plants and nesting sites (Peco et al., 2012; Lázaro et al., 2016; Segarra et al., 2023)

REFERENCES

Lázaro, A., Tscheulin, T., Devalez, J., Nakas, G., Petanidou, T., 2016. Effects of grazing intensity on pollinator abundance and diversity, and on pollination services. Ecol. Entomol. 41, 400-412.

Peco, B., Carmona, C.P., de Pablos, I., Azcárate, F.M., 2012. Effects of grazing abandonment on functional and taxonomic diversity of Mediterranean grasslands. Agriculture, Ecosystems & Environment 152, 27-32.

Segarra, J., Fernàndez-Martínez, J., Araus, J.L., 2023. Managing abandoned Mediterranean mountain landscapes: The effects of donkey grazing on biomass control and floral diversity in pastures. CATENA 233, 107503.

- Can you provide, at least as supplementary, graphical representations of the results regarding overall pollination and its relationship with the natural area amount and field management? It is very hard to follow and understand the results only with the table and the CCA results, and graphs as in Figure 2B would improve reader comprehension and give a better idea of the results, even when no differences were detected (or they were marginal).

REPLY: We think there has been a misunderstanding here. The results of overall pollinator abundance (lines 371-379) are actually those represented in Fig 2B, so we do not understand the comment. If the reviewer means that she/he would like to see the results for pollinator community composition (lines 338-357) as a linear representation, this is not possible because the analysis of community composition is a multivariate analysis (CCA) and therefore it needs to be plotted in the multivariate space, as it was done in Fig 2A.

Overall, I miss some information about the orchard management and flowering related aspects, that may be important to form conclusions and put results in perspective:

- A common p

---

## [Decision Letter · Decision Letter 1]

11 Dec 2024

PONE-D-24-27278R1Landscape conservation and orchard management influence carob tree yield through changes in pollinator communitiesPLOS ONE

Dear Dr. Lázaro,

Thank you for submitting your manuscript to PLOS ONE. After careful consideration, we feel that it has merit but does not fully meet PLOS ONE’s publication criteria as it currently stands. Therefore, we invite you to submit a revised version of the manuscript that addresses the points raised during the review process.

**The three reviewers and myself acknowledge the effort in revising this manuscript. The study has significantly improved during the revision process and it is now almost ready for its publication in Plos One. Please address the very few comments provided by the reviewers.**

We look forward to receiving your revised manuscript.

Kind regards,

Vicente Martínez López

Academic Editor

PLOS ONE

**Journal Requirements:**

Reviewers' comments:

Reviewer's Responses to Questions

**Comments to the Author**

1. If the authors have adequately addressed your comments raised in a previous round of review and you feel that this manuscript is now acceptable for publication, you may indicate that here to bypass the “Comments to the Author” section, enter your conflict of interest statement in the “Confidential to Editor” section, and submit your "Accept" recommendation.

Reviewer #1: All comments have been addressed

Reviewer #2: All comments have been addressed

Reviewer #3: (No Response)

2. Is the manuscript technically sound, and do the data support the conclusions?

Reviewer #1: Yes

Reviewer #2: Yes

Reviewer #3: Yes

3. Has the statistical analysis been performed appropriately and rigorously? 

Reviewer #1: Yes

Reviewer #2: N/A

Reviewer #3: Yes

4. Have the authors made all data underlying the findings in their manuscript fully available?

Reviewer #1: Yes

Reviewer #2: Yes

Reviewer #3: Yes

5. Is the manuscript presented in an intelligible fashion and written in standard English?

Reviewer #1: Yes

Reviewer #2: Yes

Reviewer #3: Yes

6. Review Comments to the Author

**Reviewer #1:**  I appreciate the authors’ timely response to my previous comments and their diligent efforts in revising the manuscript. The revised version addresses the concerns raised, demonstrating a strong grasp of the subject matter. The study effectively investigates the relationships between natural habitat, pollinator abundance, and sex ratios in carob tree seed production, which is of significant ecological and agricultural importance.

Overall, I find this revised article to be well-structured, informative, and valuable to the field of agricultural ecology. I recommend its acceptance for publication, as it contributes significantly to our understanding of factors influencing carob seed production and offers insights that could inform future agricultural practices.

**Reviewer #2:**  I appreciate the revision done by the authors, and all my comments were taken into account or explained/counter discussed. I have minor comments to give upon the new read of the document, but I think the manuscript is ready for publication. Congrats on the work.

Line 72 – 79: Here, I miss the indication that equilibrated sex ratios are also needed so that suitable levels of pollen (from the males) are available for fertilization of the female flowers (maybe it could be in the line before, 71-72). I know it is obvious for researchers with knowledge in this type of crop, but it would be nice the be stated.

Line 100-102: Can you revise this phrase, please? It becomes confusing the influence of the effects on pollinators. Maybe it could be improved.

Line 103-104: “Understanding how the landscape context and local management of orchards affect carob tree yields …”. This is a repetition of the previous phrase, maybe rephrase to avoid repetition.

Line 133: I suggest 1.50 km to maintain the number of decimals.

Line 214: In/For both study years.

Line 312: See Figures.

Line 315: I suggest naming honeybees, once it was the name appointed in the methods section on line 205.

Line 328-329: Similar here to the other flies.

**Reviewer #3: ** I want to thank the authors for their efforts in revising their paper, they have done an excellent job incorporating the comments of all the reviewers to produce a significantly improved manuscript.

I am satisfied with the additional information and explanation added about honey bee hives on Mallorca, and thank the authors for providing the further information! I also thank them for conducting the additional spatial autocorrelation analysis, which has strengthened the validity of their results. I also appreciate the inclusion of photos of the inflorescences as recommended by another reviewer, it’s nice have a visual of the system (as someone who’s never seen a carob tree!)

I only have a very few small grammatical edits I noticed (see below), otherwise I recommend this paper for publication upon making these last small changes.

Line 67: It’s very unusual to start a paragraph with “besides,” I would recommend taking it out and adding “also” later in the sentence (and making effect plural): “The effects of pollinator loss on crops might also depend[…]”

Line 83: Needs slight rewording to make sense, my suggestion would be: “[…] clinical applications, with the seeds being the most profitable […]”

Line 100: This should either say “might lead to spatial isolation” or “might drive spatial isolation”

Line 147: small typo, should say “independent of whether”

Line 170: add “and” after the comma (“and the male-to-female ratio”)

Line 227: Should be “weighing” (not weighting)

Line 250: Should be past tense “ran” (instead of run)

Line 533: Using “On the contrary” is usually used to negate a statement in the sentence immediately before, which was not your intention here (I think you mean to contrast the statement two sentences back) - I would recommend changing to “However,” or “Contrary to our prediction, […]” instead.

7. PLOS authors have the option to publish the peer review history of their article (what does this mean? ). If published, this will include your full peer review and any attached files.

**Do you want your identity to be public for this peer review?** For information about this choice, including consent withdrawal, please see our Privacy Policy .

Reviewer #1: No

Reviewer #2: No

Reviewer #3: **Yes: ** Annika Salzberg

---

## [Author Response · Author response to Decision Letter 1]

19 Dec 2024

Reviewer #1

I appreciate the authors’ timely response to my previous comments and their diligent efforts in revising the manuscript. The revised version addresses the concerns raised, demonstrating a strong grasp of the subject matter. The study effectively investigates the relationships between natural habitat, pollinator abundance, and sex ratios in carob tree seed production, which is of significant ecological and agricultural importance.

Overall, I find this revised article to be well-structured, informative, and valuable to the field of agricultural ecology. I recommend its acceptance for publication, as it contributes significantly to our understanding of factors influencing carob seed production and offers insights that could inform future agricultural practices.

REPLY: Thank you for your positive assessment of the manuscript.

Reviewer #2

I appreciate the revision done by the authors, and all my comments were taken into account or explained/counter discussed. I have minor comments to give upon the new read of the document, but I think the manuscript is ready for publication. Congrats on the work.

REPLY: Thank you very much for your revision. Below you will find our replies to your comments.

Line 72 – 79: Here, I miss the indication that equilibrated sex ratios are also needed so that suitable levels of pollen (from the males) are available for fertilization of the female flowers (maybe it could be in the line before, 71-72). I know it is obvious for researchers with knowledge in this type of crop, but it would be nice the be stated.

REPLY: We added this explanation to the sentence as suggested (lines 72-75).

Line 100-102: Can you revise this phrase, please? It becomes confusing the influence of the effects on pollinators. Maybe it could be improved.

REPLY: We have rewritten this sentence according to your suggestion (lines 101-104).

Line 103-104: “Understanding how the landscape context and local management of orchards affect carob tree yields …”. This is a repetition of the previous phrase, maybe rephrase to avoid repetition.

REPLY: We slightly modified the sentence to avoid repetition with the previous one (lines 104-106).

Line 133: I suggest 1.50 km to maintain the number of decimals.

REPLY: Done (line 133)

Line 214: In/For both study years.

REPLY: Added (line 214)

Line 312: See Figures.

REPLY: This is not a typo, “Ses Figueres” is the name of one of the study sites, as is “Son Cotoner” that it is named in the same sentence. We have added “” to clarify that these are names of study sites (lines 311-312)

Line 315: I suggest naming honeybees, once it was the name appointed in the methods section on line 205.

REPLY: Changed (line 315)

Line 328-329: Similar here to the other flies.

REPLY: We added “other flies” together with the description “(muscoid and small acalyptrate flies)” (line 328-329)

Reviewer #3

I want to thank the authors for their efforts in revising their paper, they have done an excellent job incorporating the comments of all the reviewers to produce a significantly improved manuscript.

I am satisfied with the additional information and explanation added about honey bee hives on Mallorca, and thank the authors for providing the further information! I also thank them for conducting the additional spatial autocorrelation analysis, which has strengthened the validity of their results. I also appreciate the inclusion of photos of the inflorescences as recommended by another reviewer, it’s nice have a visual of the system (as someone who’s never seen a carob tree!)

I only have a very few small grammatical edits I noticed (see below), otherwise I recommend this paper for publication upon making these last small changes.

REPLY: Thank you for your valuable first revision, which helped us improve the manuscript, and for your help with the grammatical edits now.

Line 67: It’s very unusual to start a paragraph with “besides,” I would recommend taking it out and adding “also” later in the sentence (and making effect plural): “The effects of pollinator loss on crops might also depend[…]”

REPLY: Done (line 67)

Line 83: Needs slight rewording to make sense, my suggestion would be: “[…] clinical applications, with the seeds being the most profitable […]”

REPLY: Done (line 84)

Line 100: This should either say “might lead to spatial isolation” or “might drive spatial isolation”

REPLY: Done (line 101)

Line 147: small typo, should say “independent of whether”

REPLY: Changed (line 147)

Line 170: add “and” after the comma (“and the male-to-female ratio”)

REPLY: Done (line 170)

Line 227: Should be “weighing” (not weighting)

REPLY: Done (line 227)

Line 250: Should be past tense “ran” (instead of run)

REPLY: Done (line 250)

Line 533: Using “On the contrary” is usually used to negate a statement in the sentence immediately before, which was not your intention here (I think you mean to contrast the statement two sentences back) - I would recommend changing to “However,” or “Contrary to our prediction, […]” instead.

REPLY: Done (line 533)

---

## [Editor Report · Decision Letter 2]

2 Jan 2025

Landscape conservation and orchard management influence carob tree yield through changes in pollinator communities

PONE-D-24-27278R2

Dear Dr. Lázaro,

We’re pleased to inform you that your manuscript has been judged scientifically suitable for publication and will be formally accepted for publication once it meets all outstanding technical requirements.

Kind regards,

Vicente Martínez López

Academic Editor

PLOS ONE
---

## [Editor Report · Acceptance letter]

PONE-D-24-27278R2

PLOS ONE

Dear Dr. Lázaro,

I'm pleased to inform you that your manuscript has been deemed suitable for publication in PLOS ONE. Congratulations! Your manuscript is now being handed over to our production team.

Kind regards,

on behalf of

Dr. Vicente Martínez López

Academic Editor

PLOS ONE